# Equipping community pharmacy workers as agents for health behaviour change: developing and testing a theory-based smoking cessation intervention

Liz Steed,[1] Ratna Sohanpal,[1] Wai-Yee James,[1] Carol Rivas,[2] Sandra Jumbe,[1] Angel Chater,[1,3] Adam Todd,[4] Elizabeth Edwards,[1] Virginia Macneil,[5] Fraser Macfarlane,[1] Trisha Greenhalgh,[5] Chris Griffiths,[1] Sandra Eldridge,[1] Stephanie Taylor,[1] Robert Walton[1]

► Prepublication history and additional material are available. To view these files please visit the journal online (http://dx.doi.org/10.1136/bmjopen-2016-015637).

[1]Asthma UK Centre for Applied Research, Centre for Primary Care and Public Health, Barts and The London School of Medicine and Dentistry, Queen Mary University of London, London, UK
[2]Faculty of Health Sciences, University of Southampton, UK
[3]University of Bedfordshire, Luton, UK
[4]Centre for Health and Inequalities Research (CHIR), University of Durham
[5]University of Oxford, Oxford, UK

**Correspondence to**
Dr Liz Steed;
e.a.steed@qmul.ac.uk

## ABSTRACT

**Objective** To develop a complex intervention for community pharmacy staff to promote uptake of smoking cessation services and to increase quit rates.

**Design** Following the Medical Research Council framework, we used a mixed-methods approach to develop, pilot and then refine the intervention.

**Methods** *Phase I*: We used information from qualitative studies in pharmacies, systematic literature reviews and the Capability, Opportunity, Motivation—Behaviour framework to inform design of the initial version of the intervention. *Phase II*: We then tested the acceptability of this intervention with smoking cessation advisers and assessed fidelity using actors who visited pharmacies posing as smokers, in a pilot study. *Phase III*: We reviewed the content and associated theory underpinning our intervention, taking account of the results of the earlier studies and a realist analysis of published literature. We then confirmed a logic model describing the intended operation of the intervention and used this model to refine the intervention and associated materials.

**Setting** Eight community pharmacies in three inner east London boroughs.

**Participants** 12 Stop Smoking Advisers.

**Intervention** Two, 150 min, skills-based training sessions focused on communication and behaviour change skills with between session practice.

**Results** The pilot study confirmed acceptability of the intervention and showed preliminary evidence of benefit; however, organisational barriers tended to limit effective operation. The pilot data and realist review pointed to additional use of Diffusion of Innovations Theory to seat the intervention in the wider organisational context.

**Conclusions** We have developed and refined an intervention to promote smoking cessation services in community pharmacies, which we now plan to evaluate in a randomised controlled trial.

**Trial registration number** UKCRN ID 18446, Pilot.

### Strengths and limitations of this study

► We used a detailed mixed-methods approach aligned with Medical Research Council recommendations to develop a theory-driven intervention to promote smoking cessation in community pharmacies.
► We used observational data, together with theories of behaviour change, to develop an initial version of the intervention which we piloted in pharmacies.
► We used standardised simulated client methods to evaluate intervention fidelity by assessing real-life pharmacy worker behaviour.
► Following piloting and feasibility testing, we refined the intervention using organisational theory to facilitate implementation.
► We developed a detailed programme theory, specified through a logic model, to operationalise the hypothesised mechanisms of change.

## INTRODUCTION

Recent years have seen increasing use of community pharmacies for health behaviour change interventions. This shift in pharmacy practice in the UK is underpinned by Department of Health policy[1] and reflected in the drive towards Healthy Living Pharmacies.[2] Perhaps the best established of the new health promotion functions is smoking cessation, and community pharmacies now have a prominent place in delivery of National Health Service (NHS) Stop Smoking Services, commissioned through a new contractual framework.[3] While there is considerable potential for community pharmacies to support people in quitting, translation into effective practice may be less than optimal, since 4-week quit rates are only 48% which is below the target 70%.[4 5] An important barrier may be a lack of awareness about the public health services now offered by community pharmacies.[6] Research also suggests that while pharmacy workers see a role for themselves in delivering interventions to change

health behaviours, their level of confidence is low,[7] possibly related to lack of training,[8] and this may limit their effectiveness.

A number of studies have aimed to improve smoking cessation services in pharmacies through development of training sessions for pharmacists. These studies suggest some benefits including an increased level of smoking cessation counselling,[9 10] higher quit rates,[11 12] increased cost effectiveness[13] and sustained positive effects on pharmacist consulting behaviour.[12 14] A Cochrane review also shows that pharmacists could increase smoking cessation rates by providing a counselling and record-keeping support programme for their customers.[15]

While previous interventions have shown benefits, these have not necessarily been translated into practice and their focus was primarily on the smoking cessation consultation itself rather than engagement into the stop smoking service. We therefore established the Smoking Treatment Optimisation in Pharmacies (STOP) programme, which aims to increase self-efficacy and motivation of pharmacy workers, promoting engagement of smokers into the NHS stop smoking service and optimising delivery of consultations. We aim to develop client-centred communication and behaviour change skills, thus enabling workers to facilitate health behaviour change in service users more effectively. The STOP intervention does not replace training provided by the National Centre for Smoking Cessation Training (NCSCT), but builds on and reinforces many of the behaviour change techniques learnt.[16]

In developing the STOP intervention, we aimed to build on results of previous studies by making the theoretical basis for the intervention more explicit and by taking into account the wider organisational context in which the intervention is intended to operate. This paper reports (1) the process of developing the theory-based intervention; (2) evaluating acceptability to smoking cessation advisers and assessing fidelity in a pilot trial; and (3) refinement of the intervention.

## PHASE I: INTERVENTION DEVELOPMENT
### Methods
We followed the Medical Research Council guidelines for development of complex interventions[17 18] aiming to develop a programme theory with hypothesised mechanisms and predicates. We recognised that in order to change three distinct target behaviours of smokers, namely (1) engagement in stop smoking services, (2) retention in the stop smoking service and (3) sustained cessation, we needed to bring about fundamental change in pharmacy workers' consulting behaviour.

We based our initial work on the Capability, Opportunity, Motivation—Behaviour (COM-B) model which is a behaviour change framework used to guide development of behavioural interventions.[19 20] In this model, behaviour is considered to be influenced by physical and psychological capacity to engage in an action, motivation towards the behaviour (including both conscious and habitual drivers) and by physical and social opportunity to perform the behaviour.[20] We considered each of these elements in relation to the three target behaviours to develop appropriate intervention materials and to plan training sessions.

The intervention was also informed by findings from five areas of work each described in detail below including qualitative studies in local pharmacies (interviews and conversation analysis of audio recoded consultations), literature synthesis, realist review of smoking cessation interventions in community pharmacies, theoretical modelling and piloting in pharmacies. We also secured input from an expert professional advisory group comprising pharmacists (n=2), health psychologists (n=2), general practitioners (n=3), stop smoking experts (n=2) and trialists (n=2).

## RESULTS
### Synthesis of prior literature
Our initial review of the literature on behaviour change interventions in community pharmacy[6 21 22] showed that such interventions could be successfully implemented and were most likely to be successful when supported by specific training. Further synthesis of the literature provided information on practicalities such as length of training, with shorter durations (2 hours) as effective, and potentially more acceptable, than longer training sessions.[23]

## QUALITATIVE STUDIES
Semistructured interviews with pharmacy workers about their experiences of the smoking cessation service delivery were conducted and analysed using a framework approach[17] based on the COM-B model.[24] One key finding was that cessation advisers had preconceived ideas about which smokers were likely to join or to drop out of the service and made judgements about the client's likelihood of quitting. These judgements then influenced the time advisers would spend encouraging a smoker to join the service. In part this behaviour resulted from lack of confidence in skills to engage and to motivate smokers, particularly those not actively asking for help. There was also a concern about potentially challenging interactions with some smokers, for example, smokers who particularly enjoyed smoking or those who did not see any harm in continuing to smoke. Advisers identified the need for extra training in client-centred consultation skills and for additional support to boost confidence. Some pharmacists felt that NHS remuneration for this activity was insufficient. Other pharmacy staff commented that if remuneration for pharmacists were increased, they could spend more time with smokers, which would result in more smokers joining and completing the service. There were some practical challenges such as insufficient time or resources that meant that advisers did not always seize opportunities for engagement and delivery of the service.

Conversation analysis was conducted on audio recordings of 16 pairs of smokers matched on gender, ethnicity, age and smoking intensity according to whether they were successful or unsuccessful in quitting. Quantitative and qualitative thematic analysis of consultation transcripts provided an overview of the data that helped us to determine aspects of the conversation within the consultation that might be associated with quitting or not quitting. Talk about the everyday experiences of smokers was common and apparently facilitated by a lack of social distance between advisers and their clients. This 'lifeworld' talk, when it was effectively used, enabled advisers to gain a better understanding of the smoker's perspective on the quit attempt. In this way advisers could give more effective support and facilitate more appropriate strategies to facilitate cessation. Correspondingly, there was better alignment in reasons for quitting and relapsing between the advisers and smokers who successfully stopped smoking compared with those who did not. It was apparent from the analysis that patient-centred talk was particularly likely to be omitted in deference to the 'voice of medicine' in non-quitters.[25] For example, the advisers would give detail about the physiology of smoking rather than addressing the social context of the smoking behaviour. This form of communication led to misalignment of adviser motivational strategies with smokers' reasons for quitting. Thus advisers often attempted to motivate clients with talk about financial savings, whereas smokers voiced health concerns more frequently as reasons for stopping smoking.

### Developing a theory base for the initial intervention

Having identified the core issues to address, we reviewed which theories would best meet our needs. Specifically we matched constructs that we intended to target with theories that targeted these constructs, thus ensuring our intervention programme theory was underpinned with sound behavioural theory and had a clear hypothesised mechanism of action. To address capability, we considered that Social Cognitive Theory[26] was relevant to target adviseors' attitudes about benefits of improving smoking cessation services both for themselves and the smoker (outcome expectancies). This theory was also relevant to enhancing adviser's confidence in engaging clients and delivering the smoking cessation service (self-efficacy). To address motivation, we considered that Self-Determination Theory[27] with its focus on intrinsic and extrinsic motivation would be useful, accounting for personal as well as external motivators. For example, financial rewards were seen as important to pharmacists running a small business. To address opportunity, we set an in-practice discussion of how to translate learning into the specific pharmacy context as a task to complete before the second training session.

Overall, our aim in developing the initial version of the intervention was to develop more effective behaviour change skills in the smoking cessation advisers and to encourage smoker-centred care while avoiding a didactic biomedical approach.[28]

### Expert advisory group

The group provided advice on each of the key elements of the intervention informing and overseeing the development process, both shaping the initial intervention and making adjustments to create the final version.

## INITIAL INTERVENTION

Consensus within the expert group suggested that a two-session face-to-face training programme targeting communication and behaviour change skills with homework tasks, social media support and a paper-based prompt tool should be the basis for the intervention. The social media support took the form of a Facebook page that included signposting to resources, links to videos on consulting styles and a discussion forum. Table 1 shows the content of the intervention with associated theory and behaviour change techniques.[29] Figure 1 shows the prompt tool developed as an aide memoir for use in smoking cessation consultations.

The pilot intervention was delivered jointly by a health psychologist (LS) and community pharmacist (DA) who was also a trained smoking cessation adviser and an NCSCT smoking cessation trainer. We made videos with actors demonstrating effective communication skills and ways of asking questions with scripts informed by our audio recordings of real-life consultations. We used role-plays, demonstration, brainstorming and problem-solving as teaching methods throughout the training sessions.

Where possible, the behaviour change techniques and communication skills in which advisers were being trained were modelled within the training sessions themselves. For example, advisers' personal motivations for implementing STOP were prompted in the same way that we intended them to elicit smokers' motivations to stop smoking. The training session on barriers and solutions for implementing the intervention used open-ended questions to elicit advisers' own solutions rather than simply offering solutions. These techniques mirror those that we expected the advisers to use with their clients.

## PHASE II: PILOT TESTING OF INTERVENTION ACCEPTABILITY AND FIDELITY

Piloting delivery of the initial intervention and assessment of fidelity was carried out at the same time as piloting the study procedures necessary for the cluster randomised trial that will be used to evaluate the final version of the intervention. The process of intervention development was iterative so that the theoretical modelling and the results from piloting the intervention informed further data collection and analysis.

**Table 1** Actor ratings for assessment of fidelity including display of smoking cessation materials, adviser communication skills and content of the consultation

|  | Actor 1 | | Actor 2 | |
| --- | --- | --- | --- | --- |
|  | Yes | No | Yes | No |
| NHS Stop Smoking Service poster displayed | 7 | 5 | 9 | 0 |
| Audio-visual information about the NHS Stop Smoking Service | 0 | 12 | 0 | 9 |
| Leaflets about the NHS Stop Smoking Service | 9 | 3 | 6 | 3 |
| Smoking prompts, for example, tar jar | 4 | 8 | 1 | 8 |
| Were other clients observed being asked about smoking? | 0 | 12 | 0 | 9 |
| Good body language | 6 | 6 | 6 | 3 |
| Good listening skills | 3 | 9 | 6 | 3 |
| Use open questions | 2 | 10 | 3 | 6 |
| Was topic of smoking raised | 0 | 12 | 0 | 9 |
| Was smoking raised directly | 1 | 11 | 0 | 9 |
| Was smoking raised indirectly | 0 | 12 | 0 | 9 |
| Was client told there is a smoking cessation service | 6 | 6 | 0 | 9 |
| Was service highlighted as free aside from NRT | 4 | 8 | 0 | 9 |
| Was it service highlighted as delivered by experts | 0 | 12 | 0 | 9 |
| Was it highlighted 4× higher success rate with programme | 2 | 10 | 0 | 9 |
| Asked whether client want referral to the service | 4 | 8 | 0 | 9 |
| Assess whether client ready to change | 2 | 10 | 0 | 9 |
| Close by saying door is always open | 5 | 7 | 0 | 9 |

NHS, National Health Service; NRT, Nicotine Replacement Therapy.

## METHODS
### Study design
#### Participants
Any stop smoking advisers working in the London boroughs of Tower Hamlets or City and Hackney, and in a community pharmacy allocated to the intervention arm of the pilot trial, were eligible to participate in the STOP training. All advisers had completed NCSCT level 2 training (details of the pilot trial have been published previously).[30] There were no restrictions on time elapsed since training as a stop smoking adviser. Advisers could be either pharmacists or other pharmacy workers such as counter assistants, providing they were trained to deliver stop smoking services.

### Acceptability of intervention
#### Attendance
Details on attendance at each training session and any reasons given for non-attendance were documented.

#### Qualitative assessment
All participants who consented to participate in the trial, regardless of whether they attended any training sessions, were invited to attend a semistructured interview 1 month after training. Views about the intervention and practical implementation were elicited. We also explored barriers and facilitators to attending training sessions and implementing the intervention. The inclusion of individuals who did not attend training allowed their perspectives on the barriers to attendance to be considered. Interviews were conducted by a researcher independent of intervention delivery (RS) to minimise risk of social desirability responses. A framework analysis of the data[31] was conducted by RS.

### Intervention fidelity and implementation in practice
We assessed fidelity of the intervention in terms of improving smoker engagement by using simulated clients, a method previously used for similar assessments in community pharmacy.[32] Two actors visited each pharmacy on different days, without prior knowledge of the pharmacy, although this had been agreed to at initial study consent. The actors presented different clinical scenarios designed to give an opportunity for counter staff to engage them with the smoking cessation service. A matrix was used to ensure that two different case scenarios were presented to each pharmacy. Both actors were familiar with portraying simulated patients in clinical scenarios in Objective Structured Clinical Examinations for medical students.

Each actor completed a checklist following the visit which included items related to their interaction with the adviser, for example, whether a client would like to attend the smoking cessation service, general adviser communication style and use of study materials such as tar jars and stop smoking posters. Both actors underwent training before visiting pharmacies and were blind to training status.

### Self-efficacy
An important hypothesised theoretical mechanism for change in adviser behaviour was through increased self-efficacy. We assessed this using a validated 0–5 point Likert scale, previously used in a similar context[33] such that higher ratings indicated greater self-efficacy. The scale was sent for self-completion 1 month after training to all advisers who attended.

## RESULTS
### Intervention acceptability
All 12 stop smoking advisers in the intervention pharmacies agreed to participate in the training programme and

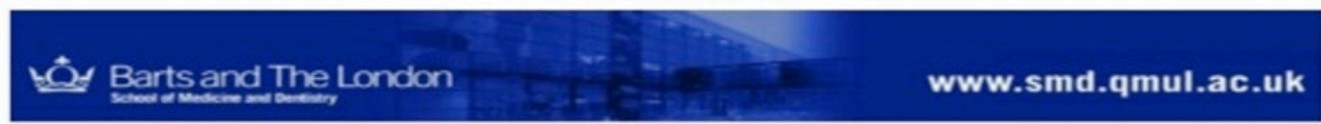

**Figure 1** The Double Whammy paper-based desk prompt. The desk prompt presents key messages from the training in a calendar format, displaying examples of how issues may be raised and addressed.

gave informed consent. In total, six advisers completed both training sessions (three at the group programme and three at a separate in-house training session). One participant withdrew after session 1 due to sickness and another because of low perceived relevance of the training sessions. Four participants did not attend any training for reasons detailed below.

In total, 8 of the 12 advisers agreed to interview, 4 of whom had attended both training sessions, 1 had attended only session 1 and 3 had not attended any training. Six of those interviewed were pharmacists, one was a counter assistant and one a pharmacy technician (age range 24–52 years). Five interviewees were male. Themes emerging from the analysis of interview transcripts that were related to the training are presented in table 1.

Logistics were identified as an important barrier to effective use of the training sessions, with participants reporting difficulty with both location (at a central training site) and timing (in the evenings 19:00 to 21:30). This was despite exploratory work that suggested evenings would be more convenient. Recommendations for venues closer to pharmacies and potential reimbursement to enable locum pharmacists to cover for pharmacist absence were made. The overall structure and content of the face-to-face training was acceptable and participants engaged well with activities, including role-play.

Only one adviser registered for the Facebook element of the intervention. Reasons for not engaging included not being allowed to use social media by managers, feeling uncomfortable with use of social media and concern related to combining social and working lives. Participants reported that they felt able to use the newly learnt skills in practice and reported using specific techniques, for example, confidence scales, to assist in consultations. Some participants, however, felt that it would take time for the new skills to become fully embedded in routine practice.

### Fidelity assessment

Actor 1 visited all 12 intervention pharmacies, but actor 2 visited only 9 pharmacies, because of limited availability. Table 2 shows ratings for the 18 items of the checklist. Actor 2 appeared to rate pharmacies more negatively than actor 1 and there was some inconsistency between responses for what might be expected to be objective items, for example, the presence of a stop smoking poster and smoking prompts such as presence of a tar jar. Few pharmacy workers with whom the actors came into contact had attended the training (approximately 10%) as they were counter assistants not specifically trained in smoking cessation.

### Self-efficacy ratings

All 11 advisers completed the pretraining questionnaire and 10 completed questionnaires post training. For pre training, median score was 4 (range 3.5–4.6) and post training 4.5 (4.0–5.0).

**Table 2** A detailed description of the Smoking Treatment Optimisation in Pharmacies (STOP) intervention showing underpinning theory and behaviour change techniques

| | Content | Theoretical basis | Behaviour change techniques |
|---|---|---|---|
| Pharmacy visit | *Explain the study to pharmacist in charge or manager. Mention potential revenue stream from smoking cessation.* Emphasise to staff how this fits well with their wider role in health promotion. Raise awareness in all staff in preparation for invitation to training. *Emphasise intervention is brief.* Address preimplementation concerns. Provide financial incentive for attending training (£30 per session, only received on completion of training). *Emphasise backing from local and national opinion leaders and organisations (eg, Local Pharmaceutical Committee, Royal Pharmaceutical Society, local CCGs and public health commissioners).* | Adoption by individuals: concerns in preadoption stage (DIT) *The innovation: compatibility; relative advantage; low complexity (DIT)* Outer context: incentives (DIT) Diffusion and dissemination: opinion leaders (DIT) | 10.2 Material incentive *(behaviour)* 9.1 Credible source 1.2 Problem Solving 6.3 Information about others' approval |
| **Training session 1** | | | |
| Introductions | General orientation to the STOP programme, information on aims of training. Include discussion of the impact of adviser behaviour on client stop smoking outcomes and health benefits to patients from stopping smoking. *Communicate the advantages of the STOP intervention over usual practice, stress it's brief and show how it fits with overall 'pharmacy' identity.* | Outcome Expectancies (SCT) *The innovation: relative advantage; compatibility; low complexity (DIT)* | 9.1 Credible source 5.1 Information on health consequences of behaviour 15.1 Verbal persuasion about capability |
| Topic 1: why are we here? | Exploration of motivation for helping smokers to quit. Does engaging and supporting smokers' quit fit with role identity, any barriers? Encourage self perception as supporters and providers of health, how one will feel if help smokers quit. Emphasise the non-medication-related, professional and public health aspects of the pharmacy role, promote a person-centred rather than product-centred ethos and foster a strong sense of professionalism. | Intrinsic and extrinsic motivators (SDT) *The innovation: compatibility (DIT)* | 5.6 Information about emotional consequences 9.2 Pros and cons 6.3 Information about others approval 13.1 Identification of self as a role model |
| Topic 2: engaging clients | Group discussion of difficult/easy clients to engage—what are potential problems, solutions? Introduction of patient-centred approach and basic communication skills including, rapport, listening and questioning. Role-play demonstration with senior pharmacist, participant practice. How to maximise opportunity with environmental resources for example, tar jars. Staff badges prompting client interaction. Addressing pharmacy workers beliefs and attitudes, for example, prejudgement of success or failure. *Emphasise predictable improved result.* | Self-efficacy (SCT) Modelling (SCT) Vicarious learning (SCT) *The innovation: relative advantage; compatibility; low complexity (DIT)* | 1.2 Problem solving 4.1 Instruction on performance of behaviour 6.1 Demonstration of behaviour 8.1 Behavioural practice and rehearsal 7.1 Prompts and cues |
| Topic 3: is this the right time? | Information on how to assess whether someone is ready to quit smoking. Using 1–10 scales to assess readiness to change and importance of change for the smoker. Role-play practice. *Does engaging and supporting smokers' quit fit with role identity?* | Self-efficacy (SCT) Self-regulation (SCT) *The innovation: compatibility (DIT)* | 4.1 Instruction how to perform behaviour 8.1 Behavioural practice and rehearsal |
| Topic 4: homework (revise NCSCT training, discuss how the intervention might be applied within your pharmacy) | Encouragement to revise NSCSCT training in smoking cessation, including quizzes. Task to discuss as a pharmacy how might implement the STOP programme within their specific pharmacy—what would be facilitators or barriers? *Emphasise testing, trialling and adaptation to local circumstances.* | Self-efficacy (SCT) Self-regulation (SCT) *The Innovation: trialabilty (DIT)* | 1.1 Goal setting *(behaviour)* 1.2 Problem solving |
| **Training session 2** | | | |

Continued

**Table 2** Continued

| | Content | Theoretical basis | Behaviour change techniques |
|---|---|---|---|
| Topic 1: feedback and reflections from homework | Discussion of homework. Key things learnt from completing NCSCT related training. Goal setting for filling in gaps. *Identification of organisational barriers, facilitators to implementing STOP in individual pharmacies. Facilitating action plans to implement STOP in their pharmacy.* *Any further thoughts on how the intervention can be adapted to local circumstances?* | Self-regulation (SCT) *The Innovation: fuzzy boundaries (DIT)* *Adoption by individuals: concerns in preadoption stage (DIT)* | 2.7 Feedback on outcome of behaviour 1.1 Goal setting 1.2 Problem solving 1.4 Action planning |
| Topic 2: challenge of changing behaviour and maintaining a client-centred stance | Brainstorming on what influences people to change behaviour—the role of beliefs, capability and opportunity in addition to knowledge. How to elicit individuals' motivations, barriers and potential strategies to change behaviour versus offering solutions. Using 'What else questions'. Understanding the 'non-smoker identity' and how to communicate to client. Demonstration and role-play. What makes this client-centred approach difficult, advantages, disadvantages, barriers and strategies to aid implementation. | Outcome expectancies (SCT) Modelling (SCT) Self-efficacy (SCT) | 4.1 Instruction on how to perform behaviour 6.1 Demonstration of behaviour 8.1 Behavioural practice and rehearsal 9.2 Pros and cons 1.2 Problem solving |
| Topic 3: planning a quit and dealing with lapses | Planning a quit—How to help people make a specific plan using a SMART approach. Setting own SMART goal. What to do if someone has a lapse, how to maintain support without excessive praise. Watch and reflect on video of strong and weak consultations of quit planning. Discussion of how to talk about willpower and the role of the open door. Demonstration and role-play. *Introduction of social media support* *Using client wallet cards to remind patients of quit plans and motivators.* | Modelling (SCT) Self-efficacy (SCT) | 4.1 Instruction on how to perform behaviour 1.1 Goal setting (behaviour) 6.1 Demonstration of behaviour 8.1 Behavioural practice and rehearsal |
| Topic 4: implementing STOP | How to translate training to practice—barriers and solutions. Use of prompts/cues including the Double Whammy (a desk top reminder with visual cues and example questions to ask) and badges *prompting client interaction.* | Self-regulation (SCT) Intrinsic/extrinsic motivators (SDT) *The innovation: augmentation/support (DIT)* | 4.1 Instruction on how to perform behaviour 1.2 Problem solving 7.1 Prompts and cues 3.1 Social support (practical and emotional) |
| Topic 5: goal setting and making a commitment | Develop a goal and elicit a commitment from participants to deliver STOP programme. Participants provided with a certificate for attending the training which is eligible for CPD points. | Modelling (SCT) Intrinsic/extrinsic motivators (SDT) *Outer context: incentives (DIT)* | 15.1 Verbal persuasion about capability 1.1 Goal setting (behaviour) 1.9 Commitment 10.2 Material reward |
| Follow-up visit | *Promote adaptation of non-core elements of the intervention through a prompted pharmacy team meeting to discuss implementation of the intervention according to the needs of each individual pharmacy for example, appointment of individual champions, monthly 'STOP' smoking days.* *Provide financial reward for those who have completed intervention training.* | *The innovation: trialability; reinvention; fuzzy boundaries; champions (DIT)* | *1.4 Action planning* *10.2 Material reward* |
| | Be delivered in mixed groups of pharmacists and other pharmacy workers to promote cohesive working practices within the individual pharmacies. | *Implementation and routinisation: organisational structure (DIT)* | *3.2 Social support (practical)* |

Features of the initial intervention are in roman text and final intervention in italics.
NCSCT, National Centre for Smoking Cessation Training; CCG, Clinical Commisioning Group; DIT, Diffusion of Innovations Theory; SCT, Social Cognitive Theory; SMART, Specific, Measurable, Attainable, Realistic, Timely; CPD, Continuing Professional Development.

## PHASE III: INTERVENTION REFINEMENT

Key issues arising from the pilot and fidelity assessments that needed to be addressed in revising the STOP intervention were

► ensuring that all pharmacy workers in contact with smokers were trained in smoker engagement (eg, all counter assistants and not just those trained as smoking cessation advisers);

► encouraging full attendance at the training sessions and display of study materials, financial reward (£30 cash per participant per session);

► understanding organisational barriers to implementation of the intervention in community pharmacies and how these might be overcome.

In response to these issues, we used realist review techniques to synthesise previous research on implementation of complex interventions for smoking cessation in community pharmacies, aiming to identify factors associated with success or failure of the intervention. The analysis took into account the environment in which the intervention was delivered and factors associated with the recipient, together with features of the intervention itself that might influence effectiveness.[34] We gained insight into how specific interventions achieved their outcomes and, where there was an intervention that was only partially effective, the review

indicated areas where aims had been achieved and the reasons why they might not have been achieved. The preliminary realist analysis identified four mechanisms on which previously reported interventions had been operating: pharmacist identity, pharmacist capability, pharmacist motivation and stakeholder confidence. These mechanisms highlight specific attributes both of pharmacists and the wider organisational context that could potentially be exploited to enhance the effectiveness of pharmacy interventions for smoking cessation (see online supplementary table S1).[35]

We selected Diffusion of Innovations Theory (as adapted by Greenhalgh *et al* for the study of organisational innovation in healthcare)[36] to seat the intervention within the organisational context of community pharmacies. This theory can be used to address how new methods of service delivery can be embedded and sustained in organisations. Amendments to the intervention according to the results of the initial piloting and review of theory are illustrated in table 3 in italics. An overall description using the TIDieR framework[37]— a tool to facilitate explicit reporting of interventions—is presented in online supplementary table S2.

The programme theory giving an overview of the intended operation of the intervention represented as a logic model[38] is shown in figure 2.

| Table 3 | Summary of qualitative findings from interviews with smoking cessation advisers related to intervention training |
| --- | --- |
| **Themes** | **Illustrative quotations** |
| Logistics/organisation of training | 'I think it was way out … I've never been there before, and because I close six thirty, … And I can't just leave at six thirty on the dot, I've got to tidy up things. So sometimes I don't leave here until maybe seven twenty or seven thirty.' |
| Suggestions to improve the logistics of training | 'For us as pharmacists it's difficult because we need to have ourselves covered by another pharmacist, otherwise the business can't run. So unless there's compensation for getting a locum pharmacists.' |
| Training structure, content including the social media support element | 'Yeah it was great … got us involved … It was very informative, the way they actually made us do some play-acting to actually show a point to why they were doing that particular part of the talk. So that was all good. to join (facebook) you would have to enrol as a company. So I don't think the boss really wanted Facebook as a company ,…' (Pharmacist) |
| Reflection on skills learnt | 'So I think this research should carry on for another year, not just six months. Because obviously we as practitioners need time … Because it's something new to us as well … There's more learning for us to do. It takes time for us to participate and engage. Like when using the scales, when I try and engage customers I'll ask them on a scale of one to ten how ready are you to give up, one being not so ready and ten being very ready. So using those trigger questions they're very good, in that little book that we were given.' |
| Improvement suggestions for training structure and content | 'I wonder whether there's an ability for the (other pharmacy) staff to be let off work maybe at a training session just for staff … or it could be a session where someone from your they can maybe …, come in for half an hour and do a similar thing to what we did. But not that in-depth that we were trained.' |
| Application of learnt skills in practice and outcomes | 'When you come up to someone, say they came in and then they don't come back for a few weeks and then come back again, it's actually to keep the people in. So all the bits we've learnt additionally that we found now we've got a better success rate because people are coming back. Because we're saying … like sometimes if you talk to someone and you say oh well, you've smoked today, you shouldn't have really done that, but now we don't say that. We say look, this is a way to try around that, let's see what else we can try to stop you getting into that situation. That sort of thing. So we're using different ideas with them.' |

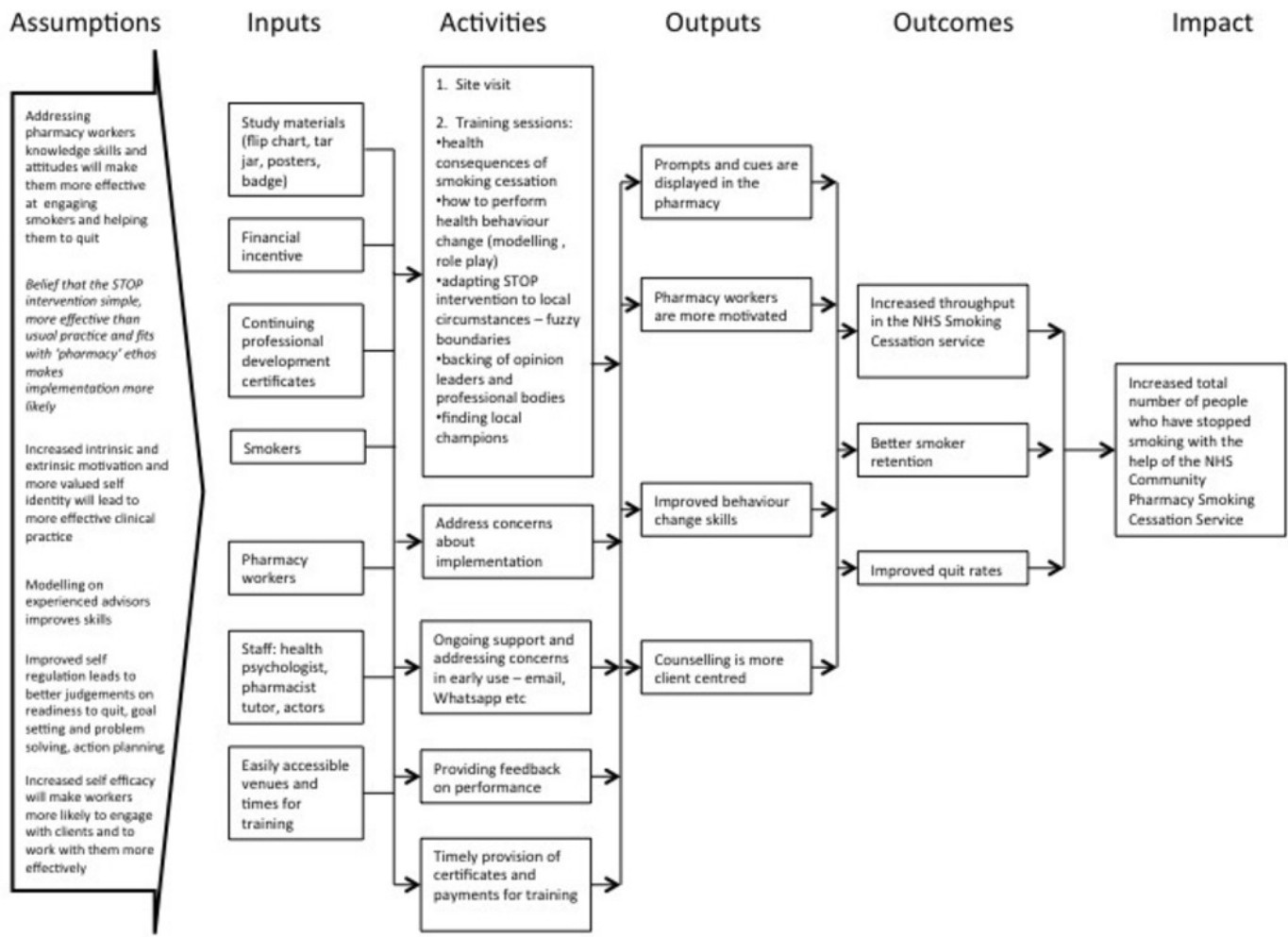

**Figure 2** Smoking Treatment Optimisation in Pharmacies programme theory. Logic model of the intended operation of the intervention.

## DISCUSSION
### Statement of principal findings
We developed the STOP intervention in line with current Medical Research Council guidance for complex interventions, incorporating information from previous literature and results of our qualitative work with cessation advisers. We drew on theories about psychological aspects of human behaviour to develop our initial intervention. Interviews with pharmacy workers after the pilot study suggested that the face-to-face training was well received by those attending and that such training might improve self-efficacy in delivering smoking cessation interventions. However, only 6 of 12 participants attended both training sessions and 4 individuals did not attend any sessions, suggesting a need for changing recruitment processes.

Our realist review[35] suggests that interventions to bring about behaviour change in pharmacy workers need to take into account role-identity, capability, motivation and the wider organisational context in which the intervention is operating. Behaviour change requires practitioner practice and continuing support with feedback[39] both to develop skills and to overcome anxieties about implementing a new form of practice. In the initial intervention, this support was anticipated to be available through social media, but given that this was not readily accepted, a more traditional approach of follow-up sessions within practice is proposed for the revised intervention.

Social media is a relatively new phenomenon and currently seen as an area of great potential for many behavioural interventions;[40] however, our study identifies possible operational barriers that need to be considered. For example, healthcare professionals may not find it acceptable to blur the boundaries between professional and social life. For this form of intervention to be successful, it may be necessary to give greater consideration to the motives and barriers towards using social media.[41] We selected 'Facebook' as one of the most well-known forms of social media and hence one we anticipated participants to be familiar with; however, it is possible that different social networking forums may have been more acceptable.

Explicit attention needs to be given to the implementation of the intervention and integration with routine practice. Previous systematic reviews in other fields have similarly highlighted the importance of implementation.[42] In the revised STOP intervention, we addressed

effective implementation using Diffusion of Innovations Theory. This theory aims to explain how new ideas spread in an organisational system and was useful in identifying key items that we needed to include in the intervention; for example, allaying concerns before adoption and demonstrating that the intervention is not complex and fits well with the expanded role of community pharmacies. We also highlighted the need for flexibility in how the training sessions are delivered, for example, at a central location or in individual pharmacies. The revised intervention training will be offered at different times of day and on different days of the week and those who are unable to attend will be offered on-site training.

## Comparison with other studies

The pilot findings from STOP are in line with previous studies that found smoking cessation training acceptable to pharmacy workers.[9 10] The finding that logistics and organisational issues may act as a barrier to implementation is also in line with evidence that potentially effective interventions for smoking cessation advisers have been developed but may not have changed quit rates or uptake of smoking cessation services.[43] The explicit application of organisational theory in development of the STOP intervention is an attempt to address this problem.

A realist review by our own team, undertaken in parallel with the fieldwork on this complex intervention and feeding into its final refinement, confirmed two key aspects of our empirical findings. First, that training in both factual knowledge and skills to enhance self-efficacy, informed by theories of behaviour change, improves the capacity and confidence (capability and motivation) of front-line pharmacy staff in supporting smoking cessation. Second, that organisational barriers (opportunity) may be significant even when individual motivation and capability are targeted. An additional finding from the realist analysis of the literature suggests that uptake and sustainability of pharmacy-based smoking cessation support depends on public and professional trust in the extended role of the pharmacist, which in turn depends on positive messages from the media and professional bodies. In the revised intervention, we recommend support from local champions such as Healthy Living Leads, which may partly address this issue.

## Strengths and weaknesses

A strength of the current study was the development work prior to piloting the intervention and using a recommended behavioural intervention development system to guide this.[19] We conducted qualitative work in community pharmacies, which helped us to gain an understanding of the detailed interactions between cessation advisers and their clients and used this to develop and refine the intervention. We concurrently developed a theoretical framework to underpin the intervention, drawing both from psychological theories related to behaviour change and organisational theory attempting to explain how interventions become incorporated into routine practice.

Having established a framework, we translated this into behaviour change techniques to facilitate change in our three target behaviours: (1) increased engagement of smokers, (2) retaining individuals in the stop smoking service and (3) improving quit rates. Our work builds on the behaviour change techniques thought to be most effective for smoking cessation[16] but which may not be well implemented in routine practice.[43]

It is a further strength of the current study that we addressed organisational aspects likely to influence the effectiveness of the interventions. Studies developing smoking cessation interventions to date have mainly focused on theories of individual behaviour change such as the transtheoretical model.[44]While studies designed to test this approach show some benefit,[45] such theories tend not to take into account broader considerations of how the intervention might be implemented. We are not aware of previous programmes in community pharmacies that have explicitly used organisational theory in addition to psychological theory to develop an intervention. Although some previous studies have compared different models of service delivery, these have not provided a theory-based exploration of the mechanisms by which the intervention brought about behaviour change in different settings.[46 47] As a result, it is difficult to generalise from these trials because mechanisms may be operating differently in different community pharmacies.[48] A consideration of theoretical perspectives on the implementation of interventions to supplement psychological theories of behaviour change would seem prerequisite if the intervention is to work reliably in diverse settings.

Our assessment of intervention fidelity is also a strength.[43] In this study, we used simulated clients to assess the degree to which core elements of the intervention were reflected in engagement behaviour of pharmacy staff. These methods have previously been used successfully in community pharmacies.[32 49 50] The simulated client methodology was informative in identifying areas where the intervention needed increased focus, for example, the need to extend training to all community pharmacy workers. While the simulated client visits provided a qualitative assessment of intervention fidelity and engagement behaviour, it was a weakness that there was no quantitative assessment due to the small scale of the pilot study. In addition, we found some discrepancy between actors in scoring apparently objective measures such as presence or absence of stop smoking posters. This raises questions over the reliability of this method and the need for increased training of actors to ensure any assessment is robust.

A weakness of the study was that both our preintervention development (qualitative studies) and piloting drew on the experience of pharmacies which were sufficiently engaged in the topic area to participate in a research study. There may be additional barriers to implementation for pharmacies that do not see smoking cessation as a priority. In addition, there may be practical problems such as lack of a private consulting room, which could

make implementation difficult in some pharmacies. The recent development of the Healthy Living Pharmacy framework[2] will mean that pharmacies are increasingly better equipped for smoking cessation and other health behaviour change tasks. It is therefore timely for pharmacy workers to be suitably trained in behaviour change skills.

A limitation of the qualitative data from the adviser interviews was that advisers' self-report of their experience of attending training sessions may be subject to social desirability responses. Nevertheless, the fact that a researcher independent from the intervention team conducted the interviews adds some validity to this assessment.

### Implications for clinical practice and policymakers

We found that smoking cessation advisers were generally motivated to support their clients to change health behaviours and were receptive to additional skills training. However, we identified a gap in referral of smokers from non-smoking cessation-trained counter assistants to trained smoking cessation advisers. This suggests that in order to engage smokers more effectively a wider work force than stop smoking advisers needs to be trained. Nevertheless, the results from our pilot studies provide initial evidence that STOP may be an acceptable and potentially effective intervention.[30]

### Implications for future research

While many health systems are expanding the role of community pharmacies into lifestyle management, there has been very little research on the effectiveness of pharmacy workers in their extended roles.[51] Since there are substantial economic implications for funding wider delivery of primary healthcare through pharmacies, health commissioners may need stronger evidence before decisions on allocation of funding are made. Such evidence will require large-scale studies to evaluate the effectiveness and cost effectiveness of changes in service delivery and further investigation into the organisational barriers to implementation.

### ADDITIONAL INFORMATION

The training and intervention materials, including the intervention manual, will be available on request from the authors following completion of the evaluation phase of the STOP programme.

**Contributors** LS led development of the initial version of the intervention and designed the fidelity assessment. RS, W-Y J, SJ, AC, CR, VM and EE recruited sites and collected and analysed qualitative and quantitative data. FM and TG conducted the realist analysis. AT and TG gave critical comments on the manuscript. LS and RW drafted the paper with input from all other authors. SJ assisted in refining the intervention. All authors have seen and approved the final manuscript.

**Funding** This study is a substudy of the NIHR-funded STOP programme on which RW is chief investigator and CG, SE, ST and TG are co-investigators. NIHR Programme grant RP-PG-0609-10181.

**Competing interests** None declared.

**Ethics approval** Ethical approval for the study was obtained from the NRES Committee South Central, Berkshire B (reference number: 13/SC/0189).

**Provenance and peer review** Not commissioned; externally peer reviewed.

**Data sharing statement** Full transcripts of the interviews (with identifying information removed) will be available, following completion of the project, on request from the study guarantor, Robert Walton, r.walton@qmul.ac.uk.

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
