## [Reviewer comments · BMJ Open]

ARTICLE DETAILS

TITLE (PROVISIONAL)	Equipping community pharmacy workers as agents for health behaviour change: developing and testing a theory-based smoking cessation intervention
AUTHORS	Steed, Liz; Sohanpal, Ratna; James, Wai Yee; Rivas, Carol; Jumbe, Sandra; Chater, Angel; Todd, Adam; Edwards, Elizabeth; MacNeill, Virginia; Macfarlane, Fraser; Greenhalgh, Trisha; Griffiths, Chris; Eldridge, Sandra; Taylor, Stephanie; Walton, Robert

VERSION 1 - REVIEW

REVIEWER	Roberto Boffi Istituto Nazionale dei Tumori, Milano
REVIEW RETURNED	18-Jan-2017

GENERAL COMMENTS	This paper is very interesting and well written. I consider these minor clarifications and modifications are necessary: - Why did the researchers decide to evaluate also pharmacists who didn't attend the training? What are your conclusions about fidelity construct?- At pag. 9 you say that "Advisors could be either pharmacists or other pharmacy workers such as counter assistants." But in the strength and weakness section, at pag. 17, you state: "our pilot training was targeted at smoking cessation advisors and did not involve the counter assistants working in the pharmacies". What does it mean?- Actor 1 evaluated 12 pharmacies, actor 2 only 9. Why this difference in number?- Why did you choose to use Facebook and not another social network? From which scientific source did you get the idea?- Were the pharmacists informed about a possible evaluation of their work with the smokers by an actor? And after the evaluation, did the debriefing take place? The pharmacists who interacted with the actors were clearly identified as trained or not trained, so it seems that their data are not anonymous.- To complete the antismoking training of pharmacists, it could be discussed the opportunity to add in the future a training on the measurement of carbon monoxide in breath, as the smoking status analysis, motivational tool and monitoring of any smoking cessation.
---

REVIEWER	Elizabeth Pogge Midwestern University United States
REVIEW RETURNED	15-Feb-2017

GENERAL COMMENTS	Overall, this manuscript is well done. I applaud the authors for this interested and novel research. I have a few minor editing suggestions; see below. Minor edits: On Page 7, line 6- "to be to be omitted"- one of the "to be" should be removed I would suggest removal of Table 1 and keep the summary of some of the qualitative finding of the study that are listed in the paragraph below it. I do not feel like this table is necessary for the article.
---

VERSION 1 – AUTHOR RESPONSE

Reviewer: 1

Reviewer Name: Roberto Boffi

Institution and Country: Istituto Nazionale dei Tumori, Milano

Please state any competing interests or state 'None declared': None declared

Please leave your comments for the authors below

This paper is very interesting and well written.

I consider these minor clarifications and modifications are necessary:

Why did the researchers decide to evaluate also pharmacists who didn't attend the training? What are your conclusions about fidelity construct?

The following sentence has been added to address why pharmacists who didn't attend the training were evaluated

'The inclusion of individuals who did not attend training allowed their perspectives on the barriers to attendance to be considered.' (pg 10)

In relation to the fidelity construct, the findings that there was limited fidelity in delivery of the intervention remain valid as the issue remains that trained staff were not interacting at the first time-point so fidelity was under threat.

At pag. 9 you say that "Advisors could be either pharmacists or other pharmacy workers such as counter assistants." But in the strength and weakness section, at pag. 17, you state: "our pilot training was targeted at smoking cessation advisors and did not involve the counter assistants working in the pharmacies". What does it mean?

We thank the reviewer for highlighting this. The confusion may come over the fact that smoking cessation advisors can be either pharmacists or other pharmacy workers but we did not included non smoking cessation trained pharmacy workers in the training. The following sentences have been amended to add clarity.

'Advisors could be either pharmacists or other pharmacy workers such as counter assistants, providing they were trained to deliver stop smoking services.' (pg 10)

'However we identified a gap in referral of smokers from non smoking cessation trained counter assistants to trained smoking cessation advisors.'

Actor 1 evaluated 12 pharmacies, actor 2 only 9. Why this difference in number?

This was purely due to practicalities as the second actor was not available to complete all 12. We have amended the text to read

'Actor one visited all twelve intervention pharmacies, but actor two visited only nine pharmacies, because of limited availability'. (pg 14)

Why did you choose to use Facebook and not another social network? From which scientific source did you get the idea?

We selected Facebook on the basis of its popularity and hence in the expectation that participants would be familiar with it's use. We have now clarified our choice and the potential for different forums to be more successful by adding the following on pg 17.

'We selected 'Facebook' as one of the most well known forms of social media and hence one we anticipated participants to be familiar with, however it is possible that different social networking forums may have been more acceptable.' (pg 17)

Were the pharmacists informed about a possible evaluation of their work with the smokers by an actor? And after the evaluation, did the debriefing take place? The pharmacists who interacted with the actors were clearly identified as trained or not trained, so it seems that their data are not anonymous.

When we first discussed the study with pharmacies the possibility of evaluation by an actor was made clear and consented to. The following has been added to make this clearer.

'Two actors visited each pharmacy on different days, without prior knowledge of the pharmacy, although this had been agreed to at initial study consent.', (pg 10)

Data were collected anonymously by the actors, who were not aware of intervention arm or individuals training status. The data were subsequently cross-referenced with intervention site to allow consideration of whether the person interacting with the actor had been trained or not. We have added the following to make this clearer.

'Both actors underwent training before visiting pharmacies and were blind to training status.' (pg 11)

To complete the antismoking training of pharmacists, it could be discussed the opportunity to add in the future a training on the measurement of carbon monoxide in breath, as the smoking status analysis, motivational tool and monitoring of any smoking cessation.

We agree this is an interesting idea and Carbon Monoxide monitoring is part of standard NHS smoking cessation service. Carbon Monoxide monitoring was not included in the current training intervention as we did not aim to replicate the standard National Centre for Smoking Cessation Training (NCSCT_ training and much of the focus was on initial engagement into the service where the acceptability of counter based assessment is less clear.

Reviewer: 2

Reviewer Name: Elizabeth Pogge

Institution and Country: Midwestern University, United States

Please state any competing interests or state 'None declared': None declared

Please leave your comments for the authors below

Overall, this manuscript is well done. I applaud the authors for this interested and novel research. I have a few minor editing suggestions; see below.

Minor edits:

On Page 7, line 6- "to be to be omitted"- one of the "to be" should be removed

Thank you, this amendment has been made.

I would suggest removal of Table 1 and keep the summary of some of the qualitative finding of the study that are listed in the paragraph below it. I do not feel like this table is necessary for the article.

Thank you for this point. We feel the table should be retained as when presenting qualitative data it is important to provide some raw data. Given the word limit of the current article it is not practical to move this all into the body of the paper. We do appreciate however that given greater space this may be preferable.

VERSION 2 – REVIEW

REVIEWER	Roberto Boffi Istituto Nazionale dei Tumori Milan Italy
REVIEW RETURNED	21-Mar-2017

GENERAL COMMENTS	The review has made the paper more clear, highlighting the limits of the research methods.
--

REVIEWER	Elizabeth Pogge Midwestern University United States
REVIEW RETURNED	15-Mar-2017

GENERAL COMMENTS	The authors have adequately addressed my concerns and I now feel the paper is acceptable for publication.
---